# Domestic Waste and Wastewaters as Potential Sources of Pharmaceuticals in Nestling White Storks (*Ciconia ciconia*)

**DOI:** 10.3390/antibiotics12030520

**Published:** 2023-03-05

**Authors:** Guillermo Blanco, Pilar Gómez-Ramírez, Silvia Espín, Pablo Sánchez-Virosta, Óscar Frías, Antonio J. García-Fernández

**Affiliations:** 1Department of Evolutionary Ecology, Museo Nacional de Ciencias Naturales, CSIC, José Gutiérrez Abascal 2, 28006 Madrid, Spain; 2Toxicology and Risk Assessment Group, Biomedical Research Institute of Murcia (IMIB-Arrixaca), University of Murcia, Campus de Espinardo, 30100 Murcia, Spain; 3Area of Toxicology, Department of Socio-Health Sciences, Faculty of Veterinary, University of Murcia, Campus de Espinardo, 30100 Murcia, Spain; 4Department of Physical, Chemical and Natural Systems, Universidad Pablo de Olavide, Carretera de Utrera, km. 1, 41013 Sevilla, Spain

**Keywords:** ecopharmacovigilance, pharmaceutical pollution, human medicines, rubbish dumps, veterinary drugs, wildlife

## Abstract

Information on the exposure of wild birds to pharmaceuticals from wastewater and urban refuse is scarce despite the enormous amount of drugs consumed and discarded by human populations. We tested for the presence of a battery of antibiotics, NSAIDs, and analgesics in the blood of white stork (*Ciconia ciconia*) nestlings in the vicinity of urban waste dumps and contaminated rivers in Madrid, central Spain. We also carried out a literature review on the occurrence and concentration of the tested compounds in other wild bird species to further evaluate possible shared exposure routes with white storks. The presence of two pharmaceutical drugs (the analgesic acetaminophen and the antibiotic marbofloxacin) out of fourteen analysed in the blood of nestlings was confirmed in 15% of individuals (*n* = 20) and in 30% of the nests (*n* = 10). The apparently low occurrence and concentration (acetaminophen: 9.45 ng mL^−1^; marbofloxacin: 7.21 ng mL^−1^) in nestlings from different nests suggests the uptake through food acquired in rubbish dumps rather than through contaminated flowing water provided by parents to offspring. As with other synthetic materials, different administration forms (tablets, capsules, and gels) of acetaminophen discarded in household waste could be accidentally ingested when parent storks forage on rubbish to provide meat scraps to their nestlings. The presence of the fluoroquinolone marbofloxacin, exclusively used in veterinary medicine, suggests exposure via consumption of meat residues of treated animals for human consumption found in rubbish dumps, as documented previously at higher concentrations in vultures consuming entire carcasses of large livestock. Control measures and ecopharmacovigilance frameworks are needed to minimize the release of pharmaceutical compounds from the human population into the environment.

## 1. Introduction

Pharmaceuticals used in human and veterinary medicine are increasingly found among environmental pollutants worldwide [1,2,3]. Sewage treatment plants are generally inefficient in removing these drugs, which are excreted as biologically active parent compounds and metabolites in the urine and faeces of medicated subjects [4,5,6]. Hence, this pollution particularly affects waters downstream of large cities and factory farms as well as soils irrigated with these waters [7,8,9]. Regulations on pharmaceutical use and waste disposal vary widely from country to country [10,11,12]. Once consumed in therapeutic treatments, or when the batches reach their expiry date, these drugs are discarded in various ways that are theoretically regulated through the application of monitoring protocols by health administrations [13,14,15]. However, ecopharmacovigilance of the safe disposal of these compounds can often be inappropriate or insufficient to prevent residues of the active compounds from reaching the environment [16,17,18]. Consequently, a large proportion of these drugs is not adequately recycled and removed, instead becoming part of the cocktail of pharmaceuticals and other pollutants in the water, soil, and biota [19,20,21].

The lethal effect of diclofenac through kidney collapse in vultures represents a paradigmatic example of the dramatic impact that pharmaceuticals can have on wildlife populations [22,23,24]. Other non-steroidal anti-inflammatory drugs (NSAIDs) safer for vultures were proposed as an alternative for livestock medication [25,26]. However, these and other avian species are not protected from exposure due to legal and illegal use in livestock and the presence of residues from legal use in human medicine [26,27,28]. In addition, there is growing concern over the occurrence of circulating levels of antibiotic mixtures in avian scavengers [29,30,31], even simultaneously with NSAIDs, in a proportion of individuals [32]. Therefore, exposure to antibiotics via carcasses of medicated livestock can lead to dysbiosis, the proliferation of opportunistic pathogens, and bacterial resistance in vultures [33,34,35]. There is much less information on the exposure of wild birds to pharmaceuticals from wastewater and urban refuse despite such contamination sources being obvious given the enormous amount of drugs consumed, excreted, and discarded by human populations. 

Once pharmaceuticals reach wastewater and urban waste dumps, their potential uptake by birds may be influenced by the frequency of foraging in contaminated flowing waters and landfills by each species [36,37]. However, there is very little information on birds’ passive ingestion of these pollutants from wastewater or mixed with food waste from the human population [8,37,38]. Indeed, distinguishing between possible incidental ingestion by birds feeding in dumps and exposure to these compounds through the consumption of contaminated food and water can be challenging. Exposure to pharmaceuticals via sewage water may be particularly frequent in the case of waterfowl or terrestrial bird species that regularly drink contaminated water [37,39,40]. In these cases, it is expected that a high occurrence of drug residues in birds may reflect indirect drug exposure. Exposure to these compounds can occur through direct ingestion of tablets or gel solutions discarded by human populations in household waste. Therefore, a low occurrence of the compounds in wild birds would be expected, as a high frequency of incidental swallowing by these species does not seem likely to occur on a regular basis.

Aside from rubbish dumps, scavenger raptors like *Milvus* kites and vultures also exploit carcasses of livestock in supplementary feeding sites (SFS) (often called “vulture restaurants”) established to favour declining populations of threatened species [41]. Several species that use both the aquatic and the terrestrial environment for foraging also exploit the feeding opportunities provided by rubbish dumps [36]. The remains consumed at these sites correspond to small fragments of meat waste discarded by the human population or remains from markets and slaughterhouses subject to variable sanitary and pollutant waste controls depending on the country. For species such as gulls that do not feed on whole carcasses of livestock in SFS, exposure to pharmaceuticals can only be understood as direct exposure to these compounds, both in landfills and through sewage. Storks may feed on fragments of carcasses of livestock or small carcasses (e.g., poultry) in vulture restaurants and other carcass dumps as well as on such remains in urban landfills. In these cases, the source of exposure may vary depending on the occurrence of these types of dumps in each region.

The white stork (*Ciconia ciconia*) experienced a marked population increase in recent decades due to the use of food resources obtained from urban waste dumps [42,43]. This means that this species greatly relies on the resources found in these sites in a large part of its area of distribution [44,45,46]. Migratory and wintering populations from central and northern Europe also make massive use of these resources during their stay in the Iberian Peninsula [47,48]. This species is considered a good indicator of environmental pollution derived from industrial activities. For example, the levels of heavy metals in nestlings are related to exposure to sources of pollution such as rubbish dumps and waste from mining and industrial operations [49,50]. It has been argued that the high frequency of antibiotic-resistant bacteria in this species is due to passive acquisition in landfills and through contaminated water [51,52] rather than direct exposure to these drugs via ingestion. This contrasts with evidence of residues of parent antibiotics and their metabolites in faeces of this species, which is attributed to foraging in landfills and bioaccumulation in their aquatic prey [53]. To our knowledge, there is no information on the presence of pharmaceuticals as circulating pollutants in this species.

Here, we evaluated the potential exposure of white storks to Active Pharmaceutical Ingredients (APIs) from medicines commonly used in human populations and livestock farming. A schematic representation of the potential sources of contamination and pathways of exposure of white storks to pharmaceuticals used in human and veterinary medicine is shown in Figure 1. We tested for the presence of a battery of antibiotics, NSAIDs, and analgesics in the blood of nestlings of this species in the vicinity of an urban waste dump in Madrid in central Spain. The use of nestlings as sentinels ensures that possible contamination occurs in the natal area through exposure to contaminated food or water. If the exposure of nestlings to these drugs arises from water provided by breeders to their nestlings, then it would be expected to find these compounds in a higher frequency of individuals because they all use water contaminated with these drugs in the study area. On the contrary, if the exposure to the APIs (hereafter pharmaceuticals) is due to the ingestion of drug tablets and gel tubes used as oral and topical medication discarded among the organic waste that forms the food provided by the parents, then it would be expected to find a low frequency in the nestlings. Each possible route of exposure can determine the concentration of the drugs as contaminant residues that reached the nestlings studied. In particular, higher concentrations would be expected in the case of ingestion of the drugs via feed than via contaminated water. In addition, both sources of exposure are possible simultaneously, which would have a multiplier effect on the concentration of drugs and on the probability of finding them in the blood of the nestlings. We carried out a literature review on the occurrence and concentration of the tested compounds in other wild bird species to further evaluate possible shared exposure routes with white storks. 

## 2. Materials and Methods

### 2.1. Study Area and Study Population

The study was carried out within the boundaries of the Southeast Regional Park of Madrid, Spain. This area is crossed by the Manzanares River after its passage through the city of Madrid. It is an area highly degraded by pollution and alteration of the habitat structure due to legal and illegal urban developments, large infrastructures, gravel extraction, and intensive irrigated agriculture and cattle ranching [54,55]. The area is very close to the largest landfill in Spain (Valdemingómez), where a large incinerator of solid waste from the city of Madrid and surrounding areas is located, as well as another landfill located in the municipal district of Pinto (Figure 2). Due to the proximity to the Madrid metropolitan area, there are several sewage treatment plants that process a portion of the sewage from this population [56,57]. In the study area, these treatment plants are located in the Manzanares River (Figure 2). After sewage treatment, the effluents are discharged into the course of the river, where it runs to the Tagus River basin [58]. The huge contribution of effluents implies that most of the Manzanares River flow consists of treated wastewater [57]. This water is used to irrigate the vegetable and corn crops present in the meadows and to flood pastures used by cattle. The sludge generated in these treatment plants is disposed of in the cultivated areas [57,59]. Due to these inputs, the area was pointed out for its high pollution levels in the soil and water [60,61]. Similarly, studies carried out on wild birds also highlighted the high levels of multiple pollutants of different origins uptaken via different routes of exposure [55,62,63].

The white stork population nesting in the study area was composed of about 500 breeding pairs nesting at high density in trees along rivers, electricity pylons, and buildings (see the distribution of nests in [64]). This population experienced very marked growth since the 1990s due to the use of landfills as a main source of food [44,47,65]. In fact, this population seems to feed almost exclusively on meat remains found in the landfills during the breeding season, including both cooked and raw human food remains, especially fragments of poultry, pork, beef, and lamb, as well as marine fish remains. Storks were also observed feeding on uncooked and packaged meat scraps, which were probably discarded by markets. This high dependence on landfills as foraging grounds is reflected in the daily trips of breeders from the nests to the landfill, the remains of domestic waste found in the nests, and the presence of hundreds of individuals observed foraging daily in the landfills [44,65]. Occasionally, adult storks may feed naturally on invertebrates and small vertebrates captured in pastures and crops in the study area. These events seem to be more related to the feeding of non-breeding adults during the breeding season, or of breeders outside the breeding season, than to the feeding of nestlings, as suggested by the low intensity and duration of this foraging activity alternating with long periods of rest in flocks.

The size of the populations using the area during migration and as a wintering area also multiplied greatly during the last decades. These populations are made up of not only local individuals and those from other Spanish regions, but also of individuals from northern and central Europe, especially from France, Switzerland, and Germany [47]. These individuals feed almost exclusively in the landfills, as indicated by their daily routine of movements from daytime resting areas to the landfills and from there to the night-time communal roosts [44,47].

### 2.2. Fieldwork

During the breeding season of 2020, we monitored a sample of stork nests located in scattered nesting colonies on the banks of the Manzanares River, approximately 3 km from the main rubbish dump (Figure 2). The nests (*n* = 10) were selected at random and accessed with a ladder in June, when the nestlings were feathered, at the age of around 40 days old and before any risk of them jumping out of the nest in the presence of the researchers. The nestlings (*n* = 20) were selected at random among those in the nests sampled, ringed, measured, and examined to assess their apparent health status. A blood sample (approximately 1 mL) was collected from the radial vein using syringes with 30G needles. Blood samples were stored in heparinized tubes and refrigerated in a portable cooler until arrival at the laboratory about two hours later, where they were kept frozen at −80 °C until pharmaceutical analysis.

### 2.3. Potential Sources of Exposure to the Pharmaceuticals

To attempt to understand the occurrence of each tested pharmaceutical (see below), a review of their use and routes of administration in veterinary and human medicine was carried out (Table 1). We assumed that human and pet medicines could end up in landfills, where they can reach the storks directly after accidental ingestion together with food. These products can also potentially be uptaken from wastewater. Apart from these routes, in the case of medicines used in production animals, their presence in meat scraps and viscera discarded by slaughterhouses, markets, restaurants, and households should also be taken into account. However, the maximum concentrations of pharmaceutical compounds in these wastes are influenced by the maximum residue levels (MRL) established for meat, viscera, and other targeted tissues, regulated in Commission Regulation (EU) No 37/2010 [66]. To assess these potential exposure routes, we consulted the databases of the Veterinary Medicine Information Centre of the Spanish Agency of Medicines and Health Products [67], which are available at https://cimavet.aemps.es/cimavet/publico/home.html (accessed on 22 August 2022), as well as those from the Online Information Centre from the Spanish Agency of Medicines and Health Products (CIMA), which is available at: https://cima.aemps.es/cima/publico/home.html (accessed on 22 August 2022). To assess the possible exposure through wastewater, we reviewed published information on the pharmacological substances and their concentration in the water from the Manzanares river (DSSTP6 sampling point from Valcarcel et al. [62]; Table 1), which crosses the nesting areas of the white storks sampled in this study.

### 2.4. Analysis of the Presence of Pharmaceuticals in Blood

Fourteen pharmaceuticals, including eleven antibiotics (nalidixic acid, tetracycline, oxytetracycline, chlortetracycline, doxycycline, marbofloxacin, enrofloxacin, ciprofloxacin, florfenicol, trimethoprim, and lincomycin), two NSAIDs (tolfenamic acid and phenylbutazone), and one analgesic (acetaminophen) were extracted and analysed with high-performance liquid chromatography and time-of-flight mass spectrometry (HPLC-TOF-MS) by following a slight modification of the technique described by Gómez-Ramírez et al. [32]. Briefly, 100 µL of whole blood was mixed with 280 µL of methanol and 10 µL of HCl (10%), and the mixture was vortexed after adding ciprofloxacin-d8 as a surrogate. Samples were then left in an ultrasound bath (Selecta) for 5 min and cooled for 5 min at −20 °C. Finally, the mix was centrifuged at 4 °C (Beckman, Microfuge-R) for 5 min at 5000× *g*. Two hundred µL of the supernatant was transferred to a vial for HPLC analysis. The detection of pharmaceuticals was conducted using an Agilent 1290 Infinity II Series HPLC (Agilent Technologies, Santa Clara, CA, USA) equipped with a multi-sample automated module and associated with a hybrid mass spectrometer with an exact mass analyser with a time-of-flight detector (TOF), specifically, an Agilent Q-TOF 6550 (Agilent Technologies, Santa Clara, CA, USA), with a JetStream Dual electrospray ionisation source and an i-Funnel. 

The experimental parameters for the HPLC and Q-TOF were set in MassHunter Workstation Data Acquisition software (Agilent Technologies, Rev. B.08.00). Standards or samples (20 µL) were thermostatted at 4 °C and injected onto an Agilent Zorbax Eclipse XDB C18 (4.6 × 150 mm, 5 µm) HPLC column at a flow rate of 0.8 mL/min. Columns were equilibrated at 40 °C. In the case of positive ionisation, solvents A (MilliQ water with 0.01% formic acid) and B (acetonitrile) were used for compound separation. For negative ionisation analysis, solvents A (MilliQ water with 5 mM ammonium acetate) and B (acetonitrile) were used. In both conditions, the elution program consisted of the following: a linear gradient from 0 to 45% solvent B in 15 min; a linear gradient from 45 to 95% solvent B in 12 min; maintenance of 95% solvent B for 3 min; the initial condition (0% solvent B) was applied for 3 min before the next injection. The total run time was 33 min. The mass spectrometer was operated in both ionisation modes. The nebulizer’s gas pressure was set to 40 psi, whereas the drying gas flow was set to 16 L min^−1^ at a temperature of 150 °C, and the sheath gas flow was set to 12 L min^−1^ at a temperature of 300 °C in both conditions. The capillary spray, nozzle, fragmentor, and octopole 1 RF Vpp voltages were 4000 V, 500 V, 350 V, and 750 V, respectively, in positive ionisation, and 4000 V, 1000 V, 360 V, and 750 V respectively, in negative ionisation. Profile data in the 100–1100 m/z range were acquired for MS scans in 2 GHz extended dynamic range mode, with 3 spectra/s, 333.3 ms/spectrum, and 2700 transients/spectrum. Reference masses at 121.0509 and 922.0098 were used for mass correction during the analysis in positive mode, whereas 112.9859 and 1033.9881 were used in negative mode. Data analyses were performed with MassHunter Qualitative Analysis Navigator software (Agilent Technologies, Rev. B.08.00). Full scan data were recorded with Agilent Mass Hunter Data Acquisition software (version B.06.00) and processed with Agilent Mass Hunter Qualitative Analysis software (version B.06.00, Service Pack 1, Agilent Technologies, Inc., Santa Clara, CA, USA, 2012). 

To identify the compounds, the retention times of the analytes were compared to a standard compound (±0.5 min) using a mix of each of the pharmaceuticals (purchased from Sigma-Aldrich, Merck KGaA, Darmstad, Germany) in methanol. The differences between the theoretical exact mass and the measured accurate masses of the analyte were ≤5 ppm. Recoveries calculated using spiked whole blood were above 100%, except for nalidixic acid (57%), acetaminophen (39%), and phenylbutazone (59%). These recovery percentages were not used to calculate the final concentration. Limits of quantification ranged from 5–10 ng mL^−1^.

## 3. Results and Discussion

There is a lack of information on the exposure of terrestrial vertebrates to pharmaceuticals used in human and veterinary medicine (Table 2). In this study, we confirmed the presence of two pharmaceutical drugs out of fourteen analysed in the blood of nestling white storks (Table 2) born in a highly polluted area due to the proximity of a large human population in the city of Madrid and its surroundings. This stork population relies on domestic waste dumps. In total, pharmaceutical traces were found in 15% of the nestlings and in 30% of the nests in a snapshot sampling in a single breeding season. Positive samples for these compounds showed circulating levels that could be quantified (Table 2).

The detected compounds are among the most commonly used in human and veterinary medicine. Acetaminophen (often sold under the brand name paracetamol) was found in the blood of two nestlings from different nests, the readings of which represent 10% of nestlings and 20% of nests. This analgesic and antipyretic drug was previously found in wastewater in the study area [61] as well as in studies conducted on river water in other regions worldwide [37]. Marbofloxacin, an antibiotic of the fluoroquinolone family used exclusively in veterinary medicine, was detected in a single nestling from a different nest. Residues of this antibiotic were found in wastewater downstream of intensive livestock farms [71,73]. To our knowledge, this study is the first to search for and find these drugs in the blood of nestlings of white storks, a species that is a facultative scavenger and predator foraging in aquatic and terrestrial environments that increasingly relies on meat waste from households discarded in landfills.

The relatively low frequency of acetaminophen and marbofloxacin in nestlings studied in our snapshot sampling suggests a relatively infrequent exposure to these drugs. Because the white stork nestlings are supplied by their parents with water from local rivers and streams, our hypothesis predicts that all nestlings should be exposed to a greater or lesser extent to this type of contamination. The Manzanares River flowing through the study area presents relatively high levels of many pharmaceutical compounds used by the human population of the city of Madrid [57,61]. Therefore, this hypothesis also predicts that other compounds should frequently be present but at low or variable levels, depending on their concentration in the flowing water. However, the apparently low frequency of these drugs, and their detection in nestlings from different nests, suggests a more likely uptake via the food provided by parents to offspring. Because, in the study area, this species obtains the vast majority of its food from rubbish dumps, it is possible that the drug residues found derive from the ingestion of the compounds discarded in household rubbish from urban areas. At least in the case of acetaminophen, this hypothesis seems feasible due to the widespread use of this painkiller by the human population, which may mean that a proportion of unused medicines in their different administration forms (tablets, capsules, and gels) are discarded with other residues in household waste [74,75,76]. Due to the foraging habits of white storks, accidental ingestion of artificial objects is possible, as they are mixed with the household waste that makes up their diet (meat scraps from poultry, cows, lamb, and pigs, as well as marine fish) in the study area and in other areas [65,77]. In fact, plastics, rubber bands, and glass fragments are documented to be swallowed passively or due to being confused with food and discarded via regurgitation in pellets [78,79]. If drug ingestion occurs via water, then we would expect siblings to show shared occurrence at similar levels during our snapshot sampling, assuming that all nestlings in a particular nest were provided with water at the same time (authors’ pers. obs.). In contrast, if exposure occurs via feed, then the presence of drug residues in the nestlings may depend on the portions of feed ingested by each of them competing for food provided by their parents. Therefore, it seems feasible to assume that siblings from the same nest may show different exposure patterns to acetaminophen if food-borne contamination occurs through accidental ingestion of medicines discarded in the rubbish, rather than via flowing water. This is supported by the lower occurrence but higher levels in nestlings positive for this drug compared to values found in osprey (*Pandion haliaetus*) nestlings feeding on contaminated fish in the Delaware River and Bay in the eastern United States [72].

The presence of the antibiotic marbofloxacin suggests an alternative explanation, as this fluoroquinolone is used only in domestic animals, and it is banned from human use. Unlike acetaminophen, which is generally administered orally in the human population, marbofloxacin is often dispensed in an injectable form in treated animals [67,80]. Therefore, several possibilities can explain the detection of the compound in the blood of a nestling stork. Previous studies found high frequencies of this and other fluoroquinolones in the plasma of several species of avian scavengers that feed on carcasses of medicated livestock available in vulture restaurants (Table 2). The concentration found in the single white stork nestling positive to this antibiotic (7.21 ng mL^−1^) was much lower than the mean values found in *Gyps fulvus* but similar to the levels in *Neophron percnopterus* (Table 2). This exposure pathway could apply to our study, considering that storks may feed on slaughterhouse or market meat leftovers present in the dumpsites but not on the entire carcasses of large livestock exploited by vultures. Thus, residues of this antibiotic could be present in concentration below MRL (ranging between 50 and 150 ng/g^−1^) in meat and viscera scraps from bovine and porcine sources for human consumption [66]. This antibiotic is included in group B of the National Plan for Residue Investigation (PNIR) in animals and fresh meats in Spain (Real Decreto 1749/1998 [81]), which implies that its presence is authorized in meat for human consumption when the levels are below the established MRL. According to PNIR, only a small number of meat and viscera samples are randomly collected annually in slaughterhouses for antibiotic analysis. For example, in 2021, from a total of 2.5 million calves and 53.8 million pigs slaughtered in Spain, only 2019 (0.08%) and 8293 (<0.01%) meat samples of beef and pork, respectively, were collected for antibiotic residue analysis [82]. This suggests that sanitary controls carried out on products for human consumption sometimes miss some consignments with the presence of active drugs at variable levels [83], which could reach human populations, household waste, and wildlife foraging in rubbish dumps. As mentioned previously for acetaminophen, the alternative that suggests this antibiotic could reach the stork nestlings via flowing water is unlikely, as this would imply finding it in a higher frequency of individuals, especially among sibling nestlings. Furthermore, in the study area or upstream of the city of Madrid, there are no industrial farms that intensively use this antibiotic (e.g., poultry or pig farms). The nestlings sampled, however, belong to nests located in an area used by cattle, which could be the source of this antibiotic through their urine in pastures flooded with river water (Figure 2), often forming large pools used by storks (authors’ pers. obs.)

This study can be considered preliminary due to the limitation in the number of compounds screened among those potentially present as local pollutants, especially from the human population. Due to logistical and economic constraints, we gave priority to increasing the number of compounds sought, considering that the number of individuals sampled (*n* = 20) is relatively high, to determine whether the presence of each pharmaceutical can at least be considered high, moderate, or null. It should be noted that in the sample of randomly selected individuals, finding no residues of many of the compounds tested means that exposure to them can be considered to be a low, and possibly null, occurrence. In this sense, the studies with “negative” results (in this case, the apparent absence in the white storks of many of the compounds sought) are as relevant as those in which contamination is shown to be much more prevalent. Of course, other compounds potentially present among the many used in human and veterinary medicine should be tested in future studies of exposure to sewage and domestic waste in this and other bird species. For example, diclofenac, one of the most frequently used NSAIDs in human medicine, was detected in local and worldwide flowing waters [84,85]. Due to the relatively short half-lives of acetaminophen (0.45–4 h, [86]) and marbofloxacin (1.6–15 h, [87,88]) depending on livestock species, the use of other types of samples such as faeces or feathers may also be useful to assess exposure in wild birds. Indeed, antibiotics (including enrofloxacin and ciprofloxacin) and their metabolites were detected at high frequency in the faeces of white storks and two gull species wintering in Doñana National Park and surrounding areas in Andalusia, South Spain [53]. In addition, recently, human and veterinary pharmaceuticals (NSAIDs and antidepressants) were detected in the feathers of fledging Mediterranean gulls (*Ichtyaetus melanocephalus*) and Sandwich terns (*Thalasseus sandvicensis*) from the Venice Lagoon (Italy), with diclofenac being the most commonly detected drug [40].

The impact of these drugs on the health of storks remains unknown. Our sampling did not detect oral lesions or other apparent alterations in the health of the analysed nestlings or other nestlings examined in 2020 (*n* = 12) in the study area. Antibiotics under non-therapeutic exposure conditions, or in variable but irregular doses over time, are the cause of bacterial resistance in humans, livestock, and wildlife, representing a global health concern [89]. Multiple studies reported high occurrences of antibiotic resistance genes in wildlife that are attributed to exposure to resistant bacteria derived from human activities and acquired from garbage dumps and contaminated wastewaters [90]. Our study suggests that, in addition to this mechanism, at least some of this bacterial resistance may be generated from direct exposure to antibiotics acquired as environmental pollutants in a variable and irregular manner in terms of frequency, concentration, and compound mix, as was suggested to occur in vultures [34]. Our results highlight the importance of evaluating pharmaceutical exposure as a source of bacterial resistance in white storks and other migratory birds that can spread such resistance over long distances [90].

## 4. Conclusions

In conclusion, this study indicates that pharmaceuticals used in human and veterinary medicine can reach wildlife through various pathways, including the potential ingestion of medicines discarded by the human population in urban waste as well as through sewage. These compounds may thereby add to the cocktail of pollutants in the white stork and other avian species in the study area. Mixtures of pharmaceuticals, together with other environmental pollutants, remain generally unexplored for their synergic adverse effects on the health of wild birds. Further studies are urgently needed to analyse the exposure to these products, their different absorption pathways, and their impact on wildlife health. Control measures and ecopharmacovigilance frameworks are needed to prevent or minimise as well as alert authorities to the release of pharmaceutical compounds from the human population into the environment.

## Figures and Tables

**Figure 1 antibiotics-12-00520-f001:**
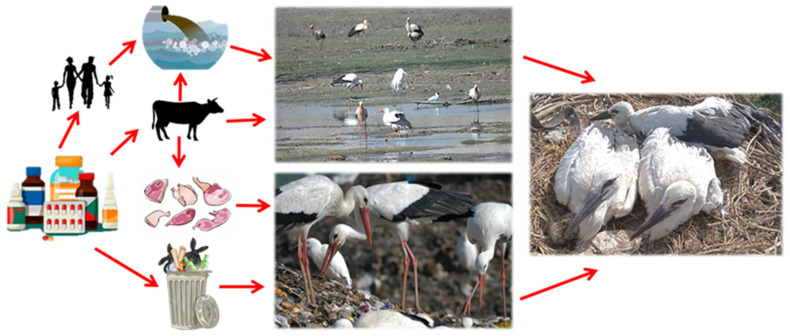
Schematic representation of the potential sources of contamination and pathways of exposure of white storks to pharmaceuticals used in human and veterinary medicine.

**Figure 2 antibiotics-12-00520-f002:**
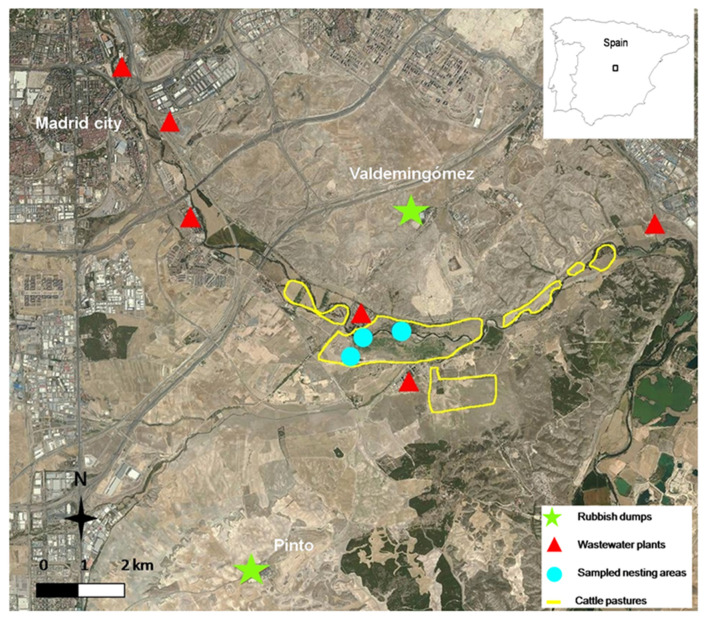
Study area showing the location of the nesting areas of the sampled white stork nestlings and the locations of the two large rubbish dumps, the sewage treatment plants, and the pastures used for cattle ranching along the Manzanares River, Madrid, Spain.

**Table 1 antibiotics-12-00520-t001:** Overview of use and potential sources of exposure to white stork nestlings in the study area. “+” refers to potential presence and “−” refers to potential lack of presence in each of the exposure sources considered.

	Use		Potential Presence in:			
Compound	Veterinary	Human	Slaughterhouse Offal ^a^	Household Waste ^b^	Wastewater (ng L^−1^) ^c^	Local Livestock
Antibiotics						
Chlortetracycline	oral (feed), topical (skin, eyes)	topical (skin, eyes)	+	+	+	+
Doxycycline	oral (feed/water), intramuscular, intravenous	oral, intravenous, periodontal	+	−	+	+
Oxytetracycline	oral (feed/water), intramuscular, intravenous, intrauterine	topical (skin, eyes, ears)	+	+	+	+
Tetracycline	oral (feed/water), intrauterine	oral	+	+	+ (<23)	+
Nalidixic acid ^d^	oral (in non-food-producing animals)	oral	-	+	+	+
Enrofloxacin	oral (feed/water), intramuscular, intravenous, subcutaneous	banned	+	−	+	+
Ciprofloxacin	banned	oral, topical (skin, eyes, ears), intravenous	+ ^e^	+	+ (<6)	+ ^e^
Marbofloxacin	oral, intramuscular, intravenous, subcutaneous, topical (ears)	banned	+	−	+	+
Trimethoprim	oral (feed/water), intramuscular, intravenous, subcutaneous	oral, intravenous, topical (ears)	+	+	+ (447)	+
Florfenicol	oral, intramuscular, subcutaneous	banned	+	−	+	+
Lincomycin	oral (feed/water), intramuscular, intravenous, breast	oral, intramuscular, intravenous	+	+	+	+
NSAIDs						
Tolfenamic acid	oral, intramuscular, intravenous, subcutaneous	banned	+	−	+	+
Phenylbutazone	oral, intravenous (in non-food-producing animals)	topical (skin)	+	+	+	+
Analgesics						
Acetaminophen	oral (water)	oral, intravenous, rectal	+	+	+	+

^a^ Refers to meat potentially reaching markets and to their residues being disposed of in household waste. ^b^ Refers to medicines (tablets, capsules, and gels) potentially discarded in household waste. ^c^ Levels in Manzanares river at DSSTP6 sampling point from Valcarcel et al. [61]. ^d^ Currently not marketed in Spain [68]. ^e^ Potentially present as a metabolite of enrofloxacin.

**Table 2 antibiotics-12-00520-t002:** Concentrations (ng mL^−1^) and frequencies of detection (%) of pharmaceuticals found in blood of white stork nestlings, and a review of occurrence in plasma and muscle in other wild birds. ND = not detected; MDL = method detection limits; LOQ = limit of quantification.

	White Storks			Other Bird Species	Reference
	% of Detection		Mean ± SD Concentration (*n*)	% of Detection (*n*), Mean or Range Concentration (*n*); Species (Year of Sampling, Country)	
Compound	In Nestlings (*n* = 20)	In Nests (*n* = 10)			
Nalidixic acid	ND	ND		3.5 (29), < LOQ; *Gyps fulvus* (2013–2015, Spain)	[32]
Tetracycline	ND	ND		3.5 (29), 1.73 (1); *Gyps fulvus* (2013–2015, Spain)	[32]
Oxytetracycline	ND	ND			
Chlortetracycline	ND	ND			
Doxycycline	ND	ND			
Marbofloxacin	5.0	10.0	7.21 (1)	72.0 (25), 62.1 (9); *Gyps fulvus* (2013, Spain) 6.3 (16), 11.5 (1); *Neophron percnopterus* (2007–2015, Spain)	[29,30]
Enrofloxacin	ND	ND		56.0 (25), 13.4 (8); *Gyps fulvus* (2013, Spain) 69.0 (29), < LOQ; *Gyps fulvus* (2013–2015, Spain) 66.0 (106), < LOQ-3.83 (17); *Gyps fulvus* (2011–2012, Spain) 100 (14), 64.1 (14); *Aegypius monachus* (2007–2015, Spain) 37.5 (16), 43.2 (5); *Neophron percnopterus* (2007–2015, Spain) 71.4 (7), 49.9 (4); *Aquila chrysaetos* (2013–2015, Spain) 0.89 (112), 1.2 (1); *Falco tinnunculus* (2018–2019, Spain) 2.7 (36), 1.2 (1); *Tyto alba* (2018–2019, Spain)	[29,30,31,32,69,70]
Ciprofloxacin	ND	ND		32.0 (25), < LOQ; *Gyps fulvus* (2013 Spain) 33.0 (106), < LOQ-0.237 (13); *Gyps fulvus* (2011–2012, Spain) 85.7 (14), 15.5 (8); *Aegypius monachus* (2007–2015, Spain) 28.6 (7), 9.2 (2); *Aquila chrysaetos* (2013–2015, Spain) not provided, 7.65 ^a^ (5); *Ardeola bacchus* (2008–2010, China)	[29,30,31,69,71]
Florfenicol	ND	ND			
Trimethoprim	ND	ND		6.9 (29), < LOQ; *Gyps fulvus* (2013–2015, Spain)	[32]
Lincomycin	ND	ND			
Tolfenamic acid	ND	ND		20.7 (29), 7.95–11.22 (6); *Gyps fulvus* (2013–2015, Spain)	[32]
Phenylbutazone	ND	ND			
Acetaminophen	10.0	20.0	9.45 ± 0.06 (2)	79.3 (29), <MDL-3.95 (23); *Pandion haliaetus* (2015, USA)	[72]
Total	15.0	30.0			

^a^ μg kg^−1^ in muscle.

## Data Availability

Data are included within the article.

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
