# Peer review of "Domestic Waste and Wastewaters as Potential Sources of Pharmaceuticals in Nestling White Storks (Ciconia ciconia)"

_antibiotics, 2023, doi:10.3390/antibiotics12030520_

Round 1
Reviewer 1 Report
Introduction
The manuscript is lacking of ecological information about the species. Authors should add some information known about diet, home range, breeding season, foraging trips, etc. Discussion could be more interesting considering the ecological context of the species. It is true that the species is large known in Europe, although people who read this kind of research could be working in other fields, and probably could ignore the characteristics of the species.
Methods
2.1. Study area and study population
You cited paper number 64, I suggest explaining the characteristics of the species nesting in the area citing this paper. It is quite uncomfortable for the lector to go to another paper to see how is the colony…if it’s considered a colony. As I mention above the manuscript could be more interesting for the lector being more explicit in the species characteristics, also considering this breeding population in particular.
Table 1 is quite interesting, simplify and resume a lot of information. I read first “2.3. Analysis of the presence of pharmaceuticals in blood” and sincerely was pretty hard to follow. I suggest presenting first “2.4. Potential sources of exposure to the analyzed pharmaceuticals” of course considering a change of subtitle. If you introduce first the pollutants adding this table, could offer a more fluent reading for the lector. The other choice is to connect the first paragraph of 2.3 with the table.
Results and Discussion
Line 278: related with the comment above in the Introduction section, the lector has not the information about the diet. Could the antibiotic be ingested by amphibians, fish or invertebrates instead of water consumption? It is good to mention in the introduction the trophic ecology of the species. This is related to line 293 and 315.
Line 349-351: I suggest to erase this paragraph. I think is the less probable of too many hypothesis mentioned by the authors.
Line 377-397: The author is finishing the Discussion with a large paragraph talking about body condition, I think the paragraph should be shortened. How about the open dump management? I think authors should mention in the final part of the discussion about landfill management in the area, for example …Is the Slaughterhouse offal deposited in the urban landfill or are deposited in other kind of landfill?
I recommend to add some literature in the Introduction:
Tsachalidis, E. P., & Goutner, V. (2002). Diet of the White Stork in Greece in relation to habitat. Waterbirds, 25(4), 417-423.
Orłowski, G., Karg, J., Jerzak, L., Bocheński, M., Profus, P., Książkiewicz-Parulska, Z., ... & Czarnecka, J. (2019). Linking land cover satellite data with dietary variation and reproductive output in an opportunistic forager: Arable land use can boost an ontogenetic trophic bottleneck in the White Stork Ciconia ciconia. Science of the Total Environment, 646, 491-502.
Gilbert, N. I., Correia, R. A., Silva, J. P., Pacheco, C., Catry, I., Atkinson, P. W., ... & Franco, A. M. (2016). Are white storks addicted to junk food? Impacts of landfill use on the movement and behaviour of resident white storks (Ciconia ciconia) from a partially migratory population. Movement Ecology, 4, 1-13.
Author Response
Introduction
The manuscript is lacking of ecological information about the species. Authors should add some information known about diet, home range, breeding season, foraging trips, etc. Discussion could be more interesting considering the ecological context of the species. It is true that the species is large known in Europe, although people who read this kind of research could be working in other fields, and probably could ignore the characteristics of the species.
Authors’ response: Thank you very much for your comments. In the new version of the manuscript we have added some details on the ecology of the species, as suggested by the reviewer. We have added this information in the 2.1. section (Study area and study population).
Methods
2.1. Study area and study population
You cited paper number 64, I suggest explaining the characteristics of the species nesting in the area citing this paper. It is quite uncomfortable for the lector to go to another paper to see how is the colony…if it’s considered a colony. As I mention above the manuscript could be more interesting for the lector being more explicit in the species characteristics, also considering this breeding population in particular.
Authors’ response: As suggested by the reviewer, we have added more details about the breeding population, indicating that it show a high nesting density, and we have specified the nesting substrates. We have added information on foraging activity and types of food consumed in the landfills, although there is no study quantifying each type of food exploited in the area.
Table 1 is quite interesting, simplify and resume a lot of information. I read first “2.3. Analysis of the presence of pharmaceuticals in blood” and sincerely was pretty hard to follow. I suggest presenting first “2.4. Potential sources of exposure to the analyzed pharmaceuticals” of course considering a change of subtitle. If you introduce first the pollutants adding this table, could offer a more fluent reading for the lector. The other choice is to connect the first paragraph of 2.3 with the table.
Authors’ response: Done.
Results and Discussion
Line 278: related with the comment above in the Introduction section, the lector has not the information about the diet. Could the antibiotic be ingested by amphibians, fish or invertebrates instead of water consumption? It is good to mention in the introduction the trophic ecology of the species. This is related to line 293 and 315.
Authors’ response: As the reviewer suggests, we have added a lines explaining in detail that the storks in the study area, especially the breeders with nestlings, feed almost exclusively on the landfills. Only perhaps non-breeding individuals could occasionally feed on invertebrates and small vertebrates, but they will always represent a very minor portion of the diet composition, and it is not expected that this type of food contains a higher prevalence of pharmaceuticals than livestock meat remains obtained from the dump.
Line 349-351: I suggest to erase this paragraph. I think is the less probable of too many hypothesis mentioned by the authors.
Authors’ response: We understand the reviewer's comment, and we agree that the hypothesis stated in the indicated sentence is the least likely in this case, but it could occur in other contexts. Since we include that possibility in the scheme of possible sources of exposure shown in Fig. 1, we would prefer to keep that sentence, unless the editor considers its elimination essential.
Line 377-397: The author is finishing the Discussion with a large paragraph talking about body condition, I think the paragraph should be shortened. How about the open dump management? I think authors should mention in the final part of the discussion about landfill management in the area, for example …Is the Slaughterhouse offal deposited in the urban landfill or are deposited in other kind of landfill?
Authors’ response: Done
I recommend to add some literature in the Introduction:
Tsachalidis, E. P., & Goutner, V. (2002). Diet of the White Stork in Greece in relation to habitat. Waterbirds, 25(4), 417-423.
Orłowski, G., Karg, J., Jerzak, L., Bocheński, M., Profus, P., Książkiewicz-Parulska, Z., ... & Czarnecka, J. (2019). Linking land cover satellite data with dietary variation and reproductive output in an opportunistic forager: Arable land use can boost an ontogenetic trophic bottleneck in the White Stork Ciconia ciconia. Science of the Total Environment, 646, 491-502.
Gilbert, N. I., Correia, R. A., Silva, J. P., Pacheco, C., Catry, I., Atkinson, P. W., Gill, J.A., Franco, A. M. (2016). Are white storks addicted to junk food? Impacts of landfill use on the movement and behaviour of resident white storks (Ciconia ciconia) from a partially migratory population. Movement Ecology, 4, 1-13.
Authors’ response: We have added the most related of the references suggested by the reviewer
Reviewer 2 Report
This paper presents Information on the exposure of wild birds to pharmaceuticals from wastewater and urban refuse. The authors tested for the presence of a battery of antibiotics, NSAIDs, and analgesics in the blood of white stork (Ciconia ciconia) nestlings in the vicinity of urban waste dumps and contaminated rivers in Madrid, Central Spain. The paper shows the importance of direct and indirect pollution way between human development and wildlife, suggesting control measures and eco-pharmacovigilance frameworks to minimize the release of pharmaceutical compounds from the human population into the environment. This paper is an excellent contribution from basic science to wildlife management and conservation of wildlife.
Author Response
Thank you very much for your positive comment on the manuscript. We are glad that you liked it and that you consider it important
Reviewer 3 Report
About this manuscript
Interesting topic with reverberance for human health since information on the exposure of wild birds to pharmaceuticals from wastewater and urban refuse is really scarce.
The authors tested for the presence of a battery of antibiotics, NSAIDs, and analgesics in the blood of white stork (Ciconia ciconia) nestlings in the vicinity of urban waste dumps and contaminated rivers in Madrid area.
Hardpoints:
The study is innovative and necessary as a One-Health study being in the topic of the Special Issue: Wildlife Sentinels of Antimicrobial Resistance
Introduction: well written, introducing elegantly the reader to the topic of this research.
M&M has been well presented and the methodology is fully replicable.
Conclusions: pointed out correctly.
References are abundant, in the frame of research, and of actuality.
What to improve:
Please introduce statistical analysis.
This will add value to this work! ANOVA is the most recommended in this case giving the p values for Potential sources of exposure (comparatively), and also a statistical correlation: Concentrations (ng mL-1 ) vs. Frequency of detection (%), which will prove the initially declared intention of the authors (in the framework of Control measures and eco-pharmacovigilance).
Author Response
Thank you very much for your positive comment on the manuscript. We are glad that you liked it and that you consider it important. Regarding the comment on the inclusion of some statistical test, we think that you are absolutely right in considering that the hypotheses should be tested using the analytical approaches that best suit the nature of the data, and the statistical testing of the hypotheses is essential in the case of experimental approximations that include different populations, conditions or control groups. However, our study is basically descriptive and, although we made hypotheses and predictions to try to explain the expected and observed results, these hypotheses cannot be statistically tested at the moment, due to the relatively small sample size and the low prevalence of pharmaceuthicals found. Future studies that deepen our hypothesis with more individuals analyzed in the study population and in others could be tested through statistical analysis, especially considering the inclusion of experimental control groups (for example, nestlings artificially fed with drug-free food).